# Sensitivity-Enhanced, Room-Temperature Detection of NH_3_ with Alkalized Ti_3_C_2_T_x_ MXene

**DOI:** 10.3390/nano14080680

**Published:** 2024-04-15

**Authors:** Yi Tan, Jinxia Xu, Qiliang Li, Wanting Zhang, Chong Lu, Xingjuan Song, Lingyun Liu, Ying Chen

**Affiliations:** 1School of Science, Hubei University of Technology, Wuhan 430068, China; 102112284@hbut.edu.cn (Y.T.); 102202245@hbut.edu.cn (W.Z.); 102312420@hbut.edu.cn (C.L.); xingjuansong@hbut.edu.cn (X.S.); liulingyun@mail.hbut.edu.cn (L.L.); chenyddc@163.com (Y.C.); 2Hubei Key Laboratory for High-Efficiency Utilization of Solar Energy and Operation Control of Energy Storage System, Hubei University of Technology, Wuhan 430068, China; 3Department of Advanced Manufacturing and Robotics, College of Engineering, Peking University, Beijing 100871, China

**Keywords:** Ti_3_C_2_T_x_ MXene, alkalization, NH_3_ sensing, room temperature

## Abstract

A layered Ti_3_C_2_T_x_ MXene structure was prepared by etching MAX-phase Ti_3_AlC_2_ with hydro-fluoric acid (HF), followed by alkalization in sodium hydroxide (NaOH) solutions of varying concentrations and for varying durations. Compared to sensors utilizing unalkalized Ti_3_C_2_T_x_, those employing alkalized Ti_3_C_2_T_x_ MXene exhibited enhanced sensitivity for NH_3_ detection at room temperature and a relative humidity of 40%. Both the concentration of NaOH and duration of alkalization significantly influenced sensor performance. Among the tested conditions, Ti_3_C_2_T_x_ MXene alkalized with a 5 M NaOH solution for 12 h exhibited optimal performance, with high response values of 100.3% and a rapid response/recovery time of 73 s and 38 s, respectively. The improved sensitivity of NH_3_ detection can be attributed to the heightened NH_3_ adsorption capability of oxygen-rich terminals obtained through the alkalization treatment. This is consistent with the observed increase in the ratio of oxygen to fluorine atoms on the surface terminations of the alkalization-treated Ti_3_C_2_T_x_. These findings suggest that the gas-sensing characteristics of Ti_3_C_2_T_x_ MXene can be finely tuned and optimized through a carefully tailored alkalization process, offering a viable approach to realizing high-performance Ti_3_C_2_T_x_ MXene gas sensors, particularly for NH_3_ sensing applications.

## 1. Introduction

Gas sensors have found widespread applications in various fields such as food safety, medical diagnosis, and hazardous gas monitoring due to their ability to track invisible environmental information [1,2,3]. To date, a variety of materials, such as nanoparticles, metal oxide semiconductors, and 2D materials, have been used to fabricate gas sensors [4,5]. Among these materials, metal oxides are extensively utilized in gas detection owing to their high sensitivity, excellent stability, and low cost [6,7,8]. Nevertheless, achieving optimal performance with metal oxide semiconductor sensors typically requires high operating temperatures, leading to substantial energy consumption. This poses a critical limitation for integrated circuits and the Internet of Things [9]. Hence, it is necessary to develop new gas-sensitive materials capable of operating at room temperature (RT).

Ammonia (NH_3_), widely used in industry, agriculture, and various other sectors, is a harmful air pollutant. NH_3_ not only threatens the stability of water ecosystems but also contributes to the formation of suspended particles in the atmosphere [10,11]. Even at low concentrations, NH_3_ can have adverse effects on human health, leading to symptoms such as vomiting, headaches, and pulmonary edema. Therefore, it is imperative to monitor the concentration of ammonia in the air. Moreover, detecting the concentration of NH_3_ in exhaled breath, generated through metabolic activities in the human body, can effectively diagnose diseases such as kidney disorders. Therefore, the development of an NH_3_ gas sensor capable of operating at room temperature under atmospheric humidity conditions with high sensitivity and a low detection limit is crucial.

MXenes, compounds consisting of transition metal carbides, nitrides, and carbonitrides, similar to graphene-like layered materials, have shown attractive applications across various fields. These applications include gas sensing, energy storage, and water purification, owing to MXenes’ excellent electronic, optical, mechanical, and thermal properties [12,13]. In particular, MXenes were reported to be promising sensing materials in gas sensors, primarily due to their high specific surface area, elevated electrical conductivity, abundant surface functional terminal groups, and efficient operation at RT [14,15]. The development of MXene-based gas sensors with high selectivity and sensitivity has attracted more and more attention as a prominent focus within gas sensor research [16,17,18]. In general, MXenes are prepared by etching the aluminum layer in MAX with HF or a mixed solution of HCl and LiF. Although the use of the mixed HCl and LiF solutions is safer in comparison to HF, MXenes prepared with this mixed solution show compact structures without significant delamination [19,20]. However, MXenes prepared with HF etching show a remarkable layered structure, making them more conducive to gas sensitivity. Additionally, the layered structure obtained with HF etching is highly stable [21,22,23]. According to previous reports, the -O and -OH surface functional terminal groups play key roles in gas detection [14,24]. However, MXenes prepared with HF etching possess a higher concentration of -F functional groups on their surface. Enhanced electrochemical performance has been achieved by reducing these -F terminal groups using various post-processing methods [25,26,27]. In this work, a simple and yet effective approach was employed to modify the functional terminal groups and expand the interlayers of Ti_3_C_2_T_x_ MXene through alkalization. Zhang et al. reported that the alkalization-grafting modification of amino-functionalized Ti_3_C_2_T_x_ MXene nanosheets yielded a highly efficient adsorption capacity for lead ions [28]. Liang et al. reported that mesoporous γ-Fe_2_O_3_ nanospheres @Ti_3_C_2_T_x_ MXene treated with LiOH showed enhanced lithium storage capabilities [29]. Bae et al. reported that the electrical properties of Ti_3_C_2_T_x_ MXene could be modified by NaOH alkalization, leading to a transition from -F terminations to -OH/-O terminations [30]. Yang et al. reported that layered Ti_3_C_2_T_x_ MXene treated with NaOH showed enhanced gas and humidity sensitivity performance owing to the increasing ratio of oxygen to fluorine functional terminal groups [31]. However, the effect of NaOH concentrations and alkalization time on the performance of NH_3_ detection has not been studied. In this work, gas sensors were fabricated based on Ti_3_C_2_T_x_ MXene alkalized with various concentrations of NaOH for different durations, and their NH_3_ gas-sensing performance at RT was investigated. Moreover, the mechanism of the sensing performance of the prepared sensors was also discussed. Our results indicate that the alkalization treatment of Ti_3_C_2_T_x_ MXene is a simple and effective method for enhancing NH_3_ gas sensing at RT. This improvement is crucial for the practical application of Ti_3_C_2_T_x_ MXene in gas sensor technology.

## 2. Materials and Methods

### 2.1. Preparation of Ti_3_C_2_T_x_ MXene

Figure 1a illustrates the schematic preparation process of Ti_3_C_2_T_x_ MXene and the alkalization of Ti_3_C_2_T_x_ MXene by NaOH. First, 1 g of Ti_3_AlC_2_ was added into a solution of 20 mL 40 wt% HF, and then continuously stirred for 24 h at 40 °C in a water bath using a magnetic stirrer. After etching, the precipitates were filtered and cleaned with deionized water by centrifugation several times (about 5 to 6 times) until reaching pH ≈ 7. Next, the residue was dried at 50 °C for 24 h in a vacuum oven. Finally, the layered-structure Ti_3_C_2_T_x_ MXene was successfully prepared.

### 2.2. Preparation of Alkalized Ti_3_C_2_T_x_ MXene

Furthermore, 0.5 g of prepared Ti_3_C_2_T_x_ MXene was added into a 30 mL 2/5/8 M NaOH solution, and then the mixture was stirred for 6, 12, 18, 24 h, respectively, at RT and RH of 40%. Subsequently, the precipitates were filtered and cleaned with deionized water by centrifugation several times until a pH ≈ 7 was achieved. The residue was then dried at 50 °C for 24 h in a vacuum oven. As a result, Ti_3_C_2_T_x_ MXene with different degrees of alkalization were successfully prepared.

### 2.3. Characterization

The morphologies of pristine and alkalized Ti_3_C_2_T_x_ MXene were characterized using field-emission scanning electron microscopy (FESEM, JEOL JSM-7500F, Tokyo, Japan) with an acceleration voltage of 15 kV. Additionally, the elemental contents of all samples were analyzed by energy-dispersive X-ray spectroscopy (EDS) attached to the SEM device. Structural characterization was performed using X-ray diffraction (XRD, Rigaku D/Max 2550, Tokyo, Japan) with Cu Kα radiation (λ = 1.5418 Å) over a scanning range of 5–80°. Elemental chemical status and compositions were determined using X-ray photoelectron spectroscopy (XPS) with Al Kα (1486.6 eV) excitation. 

### 2.4. Fabrication and Measurement of Gas Sensors

The schematic diagram of the preparation and measurement of the gas sensor is shown in Figure 1b. The prepared MXene or alkalization-treated MXene were mixed with ethanol to form a slurry, which was then applied onto an alumina ceramic tube. Subsequently, the alumina ceramic tube was dried in a vacuum oven at 50 °C for 24 h. To further enhance its structural stability, the alumina ceramic was aged in air at RT for an additional 24 h. Gas-sensing performance was evaluated using a WS-30B measuring system (Winsen Electronics Technology Co., Ltd., Zhengzhou, China) with an 18 L sensing chamber at RT and relative humidity (RH) of 40%, employing a static gas testing method. The testing gas was generated by heating a liquid into vapor. The valve formula for measuring the volume of liquid vapor was as follows:$$V_{x}=\frac{V\times C\times M}{22.4\times d\times p}\times 10^{-9}\times \frac{273+T_{r}}{273+T_{b}}$$
where *V* (mL) is the volume of the test box; *C* (ppm) is the liquid vapor concentration, in parts per million; *M* is the liquid molecular weight; *d* (g/cm^3^) is the specific gravity of the liquid, in grams per cubic centimeter; *p* is the liquid purity; *T_r_* (°C) is room temperature; and *T_b_* (°C) is the temperature in the test box.

The sensor response (Sg) can be calculated using the formula Sg = [(Rg − Ra)/Ra] × 100%, where Ra represents the resistance of the sensor in air and Rg represents the resistance of the sensor in the target gas. The response and recovery characteristics, repeatability, and stability of the sensors were evaluated under identical conditions.

## 3. Results and Discussion

Figure 1a shows a schematic diagram of the process of preparing alkalized Ti_3_C_2_T_x_ MXene. After the removal of the Al layer by HF etching, the surface of the Ti_3_C_2_T_x_ MXene is mainly composed of -F, -O, and -OH functional terminations. After alkalization, mainly -O and -OH surface groups remain with -F-rich surface terminations being transformed to -O- and -OH-rich terminations. The XRD patterns of Ti_3_AlC_2_, Ti_3_C_2_T_x_, and alkalization-treated Ti_3_C_2_T_x_ MXene using 2/5/8 M NaOH for 12 h are presented in Figure 2a,b. Figure 2c–e show the SEM images of Ti_3_AlC_2_, Ti_3_C_2_T_x_, and alkalization-treated Ti_3_C_2_T_x_ MXene using 5 M NaOH for 12 h, respectively. As shown in Figure 2a, for the Ti_3_AlC_2_, distinct peaks were observed at 2θ values of 9.52°, 19.13°, 34.03°, 36.06°, 39.8°, and 41.84°; these peaks corresponded to the (002), (004), (101), (103), (104), (105), and (110) lattice planes, respectively. But after HF etching, only (002) diffraction peak appears in the Ti_3_C_2_T_x_ MXene’ XRD pattern, the (104) diffraction peak disappears due to the removal of the Al layer, and the other peaks disappear as well [32,33]. Also, the (002) peak is shifted to a lower angle in comparison to that of the Ti_3_AlC_2_, suggesting an increase in interlayer spacing due to the extraction of the Al layer [9]. As shown in Figure 2b and the inset, after alkalization treatment, the typical diffraction peak of (002) is clearly shifted to a lower angle, from 6.35° to 5.74°, with the increase in the NaOH solution’s concentration. This indicates that the Na^+^ metal ion double stack incorporated into the MXene layer causes an increase in the lattice constant as well as the interlayer spacing [34]. With the increase in the NaOH solution’ concentration, both the number of incorporated Na^+^ metal ions and the replacement of surface terminations with -OH functional groups increase, which is clearly observed in the increasing interlayer spacing. This result is in agreement with the previous literature [30,34,35,36]. The increase in the interlayer spacing may be one of the major reasons causing the MXenes’ enhanced gas-sensing performance. In addition, the SEM results indicate that the dense layered structure of Ti_3_AlC_2_ was transformed into a layered structure due to the removal of the Al layer after being etched in the HF solution, which proves that the Ti_3_C_2_T_x_ layered structure was successfully prepared. After being alkalized in 5 M NaOH for 12 h, the alkalized Ti_3_C_2_T_x_ MXene structure shows larger stacking gaps. The SEM results are consistent with the XRD results. This layered morphology of the MXenes may play a significant role in their gas-sensing performance.

Figure 3 shows the dynamic sensing characteristics of the Ti_3_C_2_T_x_ MXene produced with different alkalization processes upon exposure to NH_3_ in a ppm concentration range of 5–500 ppm at RT and an RH of 40%. As shown in Figure 3a–c, the sensitivities of the alkalized Ti_3_C_2_T_x_ MXene sensors are clearly enhanced in comparison with those built of pristine MXene. The response magnitude of alkalized Ti_3_C_2_T_x_ MXene sensors increases with the increase in the NH_3_ concentration from 5 ppm to 500 ppm, while the sensors based on MXene undergoing 12 h of alkalization exhibit the highest response. Figure 3d shows the dynamic sensing characteristics of alkalization with 2/5/8 M NaOH for 12 h, among which the Ti_3_C_2_T_x_ MXene sensor alkalized by 5 M NaOH has the highest response; the response value of this sensor is improved by 4 to 11 times compared to the Ti_3_C_2_T_x_-based sensors, and the device achieves detection at 5 ppm. This analysis demonstrates the best combination of NaOH concentration and alkalization time for optimal sensing performance. Thus, the Ti_3_C_2_T_x_ MXene sensor alkalized with 5 M NaOH for 12 h has excellent performance and was selected for the subsequent investigation.

The response–recovery behavior is crucial for gas sensor performance. The response and recovery times can be defined and calculated based on the characteristics of response–recovery transients. Figure 4 displays the response–recovery curves of the sensors fabricated with pristine MXene and alkalized Ti_3_C_2_T_x_ MXene treated with 5 M NaOH for 12 h upon exposure to 200 ppm of NH_3_ at RT and an RH of 40%. The response time is described as the required time for the vibrational resistance to reach 90% of the stable resistance subsequent to the injecting of target gas. The recovery time is defined as the necessary time for the sensor to recover 90% of the changed resistance value in the air after releasing the target gas. As shown in Figure 4a, the gas-in switch is turned on at t = 13 s, causing an abrupt increase in sensor resistance in response to the injection of the ammonia gas. The sensor resistance is stabilized at t = 112 s, experiencing an increase from 396 Ω to 427 Ω. The response time can be characterized as the period from gas-in to the time when the resistance reaches 90% of the saturation value (425 Ω) at t = 96 s. Therefore, the response time was calculated as 96 s − 13 s = 83 s. At t = 240 s, air is introduced into the measurement, causing a decrease in resistance from 422 Ω to 402 Ω. The recovery time is described as the period from air-in to the time when the resistance decreases to 90% saturation resistance (404 Ω) at t = 324 s. Therefore, the recovery time was 324 s − 240 s = 84 s. So, the response time to ammonia gas is 83 s, and the recovery time is 84 s for the pristine Ti_3_C_2_T_x_ MXene sensor at RT and an RH of 40%. Through a similar experiment on the sensor based on Ti_3_C_2_T_x_ MXene alkalized with 5 M NaOH for 12 h, the response and recovery times were calculated as 73 s and 38 s, respectively, upon exposure to ammonia gas at RT and an RH of 40%. The results indicate that the sensors based on alkalized Ti_3_C_2_T_x_ MXene have much shorter response and recovery times compared to the sensors fabricated with pristine Ti_3_C_2_T_x_ MXene. The observed improvement in the ammonia gas response of alkalized Ti_3_C_2_T_x_ MXene sensors can be attributed to the enhanced capability to absorb and desorb ammonia gas on the surface of the alkalized Ti_3_C_2_T_x_ MXene. The details of this will be discussed later. Moreover, repeatability and stability are also important features for practical applications. Figure 4c,d show the repeatability of the sensors based on Ti_3_C_2_T_x_ MXene and Ti_3_C_2_T_x_ MXene alkalized with 5 M NaOH for 12 h in response to 200 ppm of NH_3_ for five cycles at RT and an RH of 40%, respectively. The resistance of Ti_3_C_2_T_x_ MXene and alkalized Ti_3_C_2_T_x_ MXene displayed an increase and decrease along with the change in ammonia concentration. The resistance of pristine Ti_3_C_2_T_x_ MXene did not return to the base resistance, showing base resistance drift, while the change in the resistance of the alkalized Ti_3_C_2_T_x_ MXene sensor was able to return to the base resistance; only a very small variation can be observed in sensor response, indicating a good reproducibility. As shown in Figure 4e,f, long-term stability was also observed in the sensors based on pristine Ti_3_C_2_T_x_ and Ti_3_C_2_T_x_ MXene alkalized with 5 M NaOH for 12 h upon exposure to 200 ppm of NH_3_ for about 17 days at RT and an RH of 40%, respectively. The fluctuation in the response values of the alkalized Ti_3_C_2_T_x_ MXene (30%) is smaller than that of the pristine MXene sensor (60%), which proves that alkalized MXene has good long-term stability.

In order to evaluate the performance of the alkalized Ti_3_C_2_T_x_ MXene sensor, the RT gas-sensing parameters of the present study were compared with those of other NH_3_ sensors from previous reports, as shown in Table 1. It can be observed that the gas-sensing performance in the present study is better than other listed reports in terms of the response value. Hence, Ti_3_C_2_T_x_ MXene treated with the alkalization method is a simple and effective approach to enhance the performance of NH_3_ detection at RT.

The charge transfer mechanism of the NH_3_ molecules’ adsorption/desorption process and electrical conductivity changes is shown in Figure 5. In general, Ti_3_C_2_T_x_ MXene is a p-type semiconductor wherein holes serve as the main conductive carriers, and alkalized Ti_3_C_2_T_x_ MXene also exhibits a p-type sensing behavior [16]. Therefore, the gas-sensing mechanism relies on the charge transfer during the adsorption and desorption of NH_3_ on the surface of the Ti_3_C_2_T_x_ MXene. When the Ti_3_C_2_T_x_ MXene sensor is exposed to an NH_3_ atmosphere, the NH_3_ will react with the -O and -OH surface terminals to form NH_2_, N_2_, and H_2_O and release electrons to the conduction band of the Ti_3_C_2_T_x_ MXene. The hole conductive channel can be thinned to correspondingly increase the resistance [45]. Meanwhile, when exposed to air, oxygen molecules will capture electrons from the surface of the Ti_3_C_2_T_x_ MXene, generating negative oxygen species like O_2_^−^ (at RT), resulting in the thickening of the hole conductive channel and a decrease in resistance [43,46,47]. The charge transfer process shows that -OH and -O are crucial to the adsorption of NH_3_ on the surface of Ti_3_C_2_T_x_ MXene sensors.

Therefore, the reasons for the enhanced NH_3_ sensitivity of gas sensors based on alkalized Ti_3_C_2_T_x_ MXene at RT can be explained as follows. Firstly, double-stacked Na^+^ metal ions are incorporated into the MXene layer after alkalization treatment, leading to an increase in the lattice constant and interlayer spacing. This effectively increases the specific surface area and the contact area with NH_3_, thereby significantly improving the ability of ammonia adsorption. As a result, the NH_3_ sensitivity of the alkalized Ti_3_C_2_T_x_ MXene sensor can be greatly enhanced. Zheng et al. reported that a Ti_3_C_2_T_x_ MXene treated by alkaline solution of LiOH, NaOH, and KOH showed an enhanced adsorption performance due to the expanded layer spacing and surface functional groups [48]. Wang et al. designed alkalized MXene/CoFe_2_O_4_/CS layered structures which showed ultra-strong abilities to adsorb CR, RhB, and MG dyes owing to the increased layer spacing of alkalized MXene [49]. Lian et al. also reported that alkalized Ti_3_C_2_ MXene nanoribbons were attractive for use in high-capacity ion batteries with expanding interlayer spacing [50]. Moreover, structural stability is also important for high sensing performance. The incorporated Na^+^ can effectively prevent the stacking of the Ti_3_C_2_T_x_ MXene [35]. As a result, the MXene can maintain a layered structure with high gas-sensing performance. The stability in sensor response is owed to the stable microstructure of the alkalized Ti_3_C_2_T_x_ MXene. Secondly, the -F surface functional terminations of Ti_3_C_2_T_x_ MXene are replaced by -OH and -O functional groups after NaOH alkalization, leading to a higher ratio of oxygen to fluorine atoms of the surface terminals [30,34,51]. The transformation of -F to -OH and -O are demonstrated by the XPS spectra and EDS mapping presented in Figure 6 and Figure 7. This transformation increases the number of available active sites for adsorption, thereby facilitating interactions between the MXene and ammonia, and resulting in the enhancement of the gas-sensing performance [52]. Thirdly, -OH functional groups have a negative charge, which is better able to attract positive nitrogen ions in NH_3_ molecules by electrostatic force [45], which will be discussed in detail in the next section.

The surface chemical compositions and binding state of the samples of Ti_3_C_2_T_x_ and Ti_3_C_2_T_x_ alkalized with 5 M NaOH for 12 h were investigated by XPS in detail. The survey XPS spectra of Ti_3_C_2_T_x_ and Ti_3_C_2_T_x_ alkalized with 5 M NaOH for 12 h show that the intensity of the F 1s peak becomes weaker after alkalization due to transitioning from -F to -OH terminations following alkalization, as shown in Figure 6a,e. The ratio of the number of O to F atoms is about 2.2, with an F content = 9.07 at% and an O content = 20.02 at%. After alkalization, the ratio of O to F atoms is quite high, with the F content decreasing to less than 0.1 at% and O content increasing to 24.5 at%. The increasing ratio of O to F atoms confirms that alkalized treatment can effectively reduce F and improve O, which is beneficial to the enhancement of NH_3_ gas sensing. This result is consistent with a previous report [31]. Also, the increase in O 1s also confirms that the numbers of -O or -OH functional groups are increasing on the surface of the alkalized Ti_3_C_2_T_x_. As shown in Figure 6b,f of the O 1s spectra, there are two peaks at 530.4 eV and 532.2 eV assigned to Ti-O and C-Ti-Ox bonds, respectively. Remarkably, a Ti-OH peak appears at 532.8 eV for the alkalized Ti_3_C_2_T_x_ MXene, which is not present in the pristine Ti_3_C_2_T_x_ MXene, showing the successful alkalization of the MXene. [30,53,54]. The high-resolution XPS spectra of C 1s and Ti 2p of Ti_3_C_2_T_x_ and Ti_3_C_2_T_x_ alkalized with 5 M NaOH for 12 h are displayed in Figure 6c,d,g,h, respectively. The peaks observed at 282.5 eV, 284.5 eV, 285.3 eV, and 288.2 eV are attributed to C-Ti, C-C, C-O, and C-F, respectively [37,55,56,57]. After alkalization, the C-Ti peak vanishes, and the proportion of the peak area attributed to C-O increases from 12% to 18%. This is attributed to the breaking of C atoms between Ti atoms and the bonding of C with O on the surface of the alkalized Ti_3_C_2_T_x_ [58], which indicates an increase in -OH groups in the alkalized Ti_3_C_2_T_x_ [59]. The binding energy in the Ti 2p XPS spectra is located at 455.1, 456.8, and 460.7 eV, corresponding to Ti 2p3/2, while those at 460.7, 462.1, and 464.5 eV correspond to Ti 2p1/2, which represent contributions from C-Ti^2+^(O/OH), C-Ti-F_x_, and TiO_2_ [31,54,60]. The percentage of the peak area corresponding to Ti 2p3/2 at 456.8 eV and Ti 2p1/2 at 462.1 eV from C-Ti-F_x_ of the alkalized Ti_3_C_2_T_x_ MXene significantly decreased from 9.1% to 6.6% and from 8.1% to 1.98%, respectively, in comparison to that of Ti_3_C_2_T_x_ MXene. This is attributed to the decrease in -F functional terminal groups after alkalization. On the other hand, it can be observed that the peaks of Ti 2p3/2 at 455.1 eV and Ti 2p1/2 at 460.7 eV from C-Ti^2+^(O/OH) decreased, while Ti 2p3/2 at 458.8 eV and Ti 2p1/2 at 464.5 eV from TiO_2_ increased after alkalization, indicating that Ti ions are further oxidized during the alkalization process. Overall, all the results showed an increase in the oxygen-to-fluorine atom ratio on the surface after the alkalization of Ti_3_C_2_T_x_.

Figure 7 shows the elemental EDS mappings of C, O, F, Ti, and Na and the corresponding SEM images of Ti_3_C_2_T_x_ and Ti_3_C_2_T_x_ alkalized with 5 M NaOH for 12 h. The uniform distributions of C, O, F, Ti, and Na on the surface of both pristine and alkalized Ti_3_C_2_T_x_ are evident. No Al signal is observed, indicating that the Al layers were removed in the process. The results also show the intercalation of Na^+^, the reduction in F elements, and the increase in O elements in the alkalized Ti_3_C_2_T_x_, which is in agreement with the XPS result. Figure 8 displays a statistical graph of the Na, O, F, Ti, and C element ratio and the O/F ratio of the alkalized Ti_3_C_2_T_x_ (5 M NaOH for 6, 12, 18, 24 h, 2/5/8 M NaOH for 12 h). The Ti_3_C_2_T_x_ alkalized with 5 M NaOH for 12 h shows the highest O/F ratio. It should be noted that the -OH surface functional terminal groups will become saturated if the concentration of NaOH is too high and the alkalization time is too long. This will induce negative effects on gas sensitivity. Therefore, the dynamic sensing results indicate that an optimal NaOH concentration and alkalization time can be achieved, with the best response characteristics shown by the sensors based on Ti_3_C_2_T_x_ MXene alkalized with 5 M NaOH.

## 4. Conclusions

In summary, alkalized Ti_3_C_2_T_x_ MXene were successfully prepared with different concentrations of NaOH solutions (2, 5, and 8 M) for varying duration (6, 12, 18, and 24 h). Compared to untreated Ti_3_C_2_T_x_ MXene, sensors based on the alkalized Ti_3_C_2_T_x_ MXene exhibited significantly improved gas-sensing performance, as well as enhanced repeatability and stability upon exposure to NH_3_ at RT and an RH of 40%. This improvement in ammonia-sensing capability can be attributed to the incorporation of Na^+^ metal ions into the MXene layer, leading to an increase in the specific surface area of each later and in the contact area with NH_3_. Consequently, such structural modification further enhanced the adsorption and desorption capacity for ammonia. Additionally, the introduction of -OH groups during alkalization led to an increase in the oxygen/fluorine atom ratio. Furthermore, an optimal NaOH concentration and alkalization duration were identified, with Ti_3_C_2_T_x_ MXene sensors alkalized by 5 M NaOH exhibiting the highest response in this study. Our findings underscore the effectiveness of the alkalization method in enhancing the performance of NH_3_ detection at RT and an RH of 40%, thus facilitating the practical application of Ti_3_C_2_T_x_ MXene in gas sensors. 

## Figures and Tables

**Figure 1 nanomaterials-14-00680-f001:**
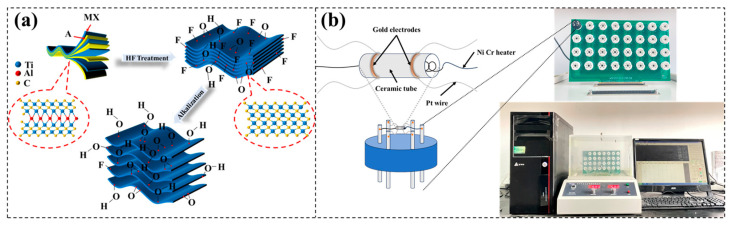
(**a**) Schematic diagram of the preparation process of alkalized Ti_3_C_2_T_x_ MXene. (**b**) Schematic diagram of the preparation and measurement of gas sensors.

**Figure 2 nanomaterials-14-00680-f002:**
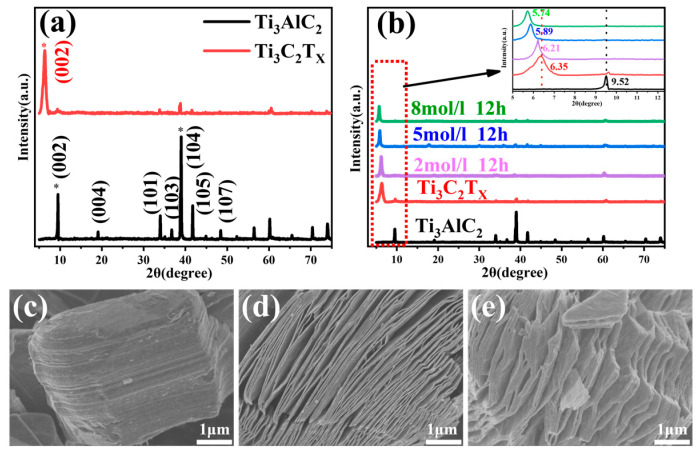
XRD patterns of (**a**) Ti_3_AlC_2_ and Ti_3_C_2_T_x_; (**b**) Ti_3_C_2_T_x_ MXene alkalized with 2/5/8 M NaOH for 12 h. SEM images of (**c**) Ti_3_AlC_2_; (**d**) Ti_3_C_2_T_x_; and (**e**) Ti_3_C_2_T_x_ MXene alkalized with 5 M NaOH for 12 h.

**Figure 3 nanomaterials-14-00680-f003:**
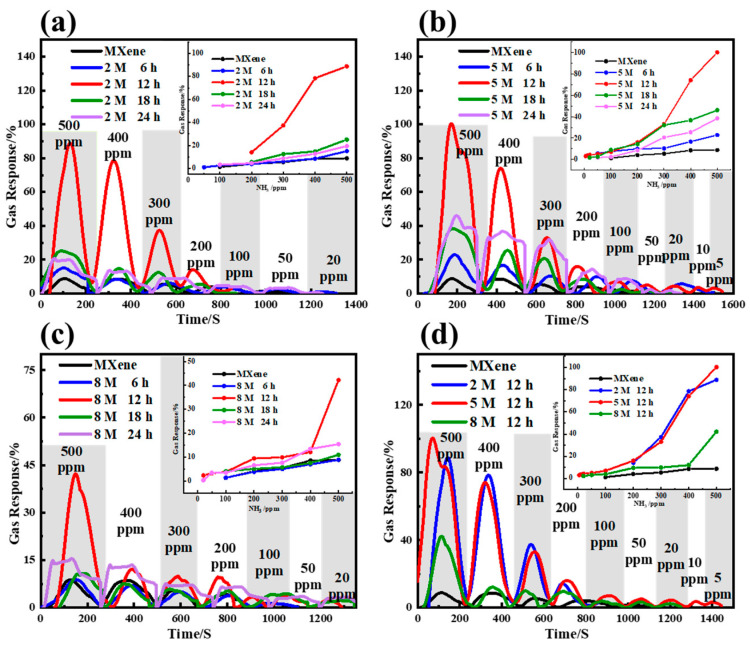
Dynamic sensing characteristics of the Ti_3_C_2_T_x_ MXene with different degrees of alkalization upon exposure to NH_3_ in ppm concentration range of 5–500 ppm at RT and RH of 40%: (**a**) 2 M NaOH for 6 h, 12 h, 18 h, 24 h; (**b**) 5 M NaOH for 6 h, 12 h, 18 h, 24 h; (**c**) 8 M NaOH for 6 h, 12 h, 18 h, 24 h; (**d**) 2/5/8 M NaOH for 12 h.

**Figure 4 nanomaterials-14-00680-f004:**
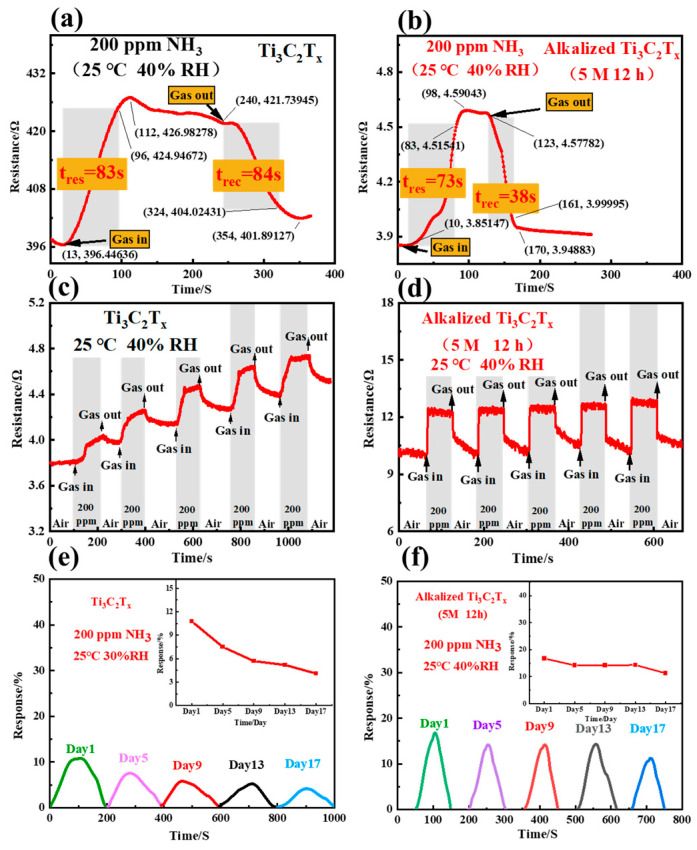
Response–recovery curves of (**a**) pristine Ti_3_C_2_T_x_ MXene sensor and (**b**) Ti_3_C_2_T_x_ MXene sensor alkalized with 5 M NaOH for 12 h exposed to 200 ppm of NH_3_ at RT and RH of 40%; (**c**) the repeatability of Ti_3_C_2_T_x_ MXene sensor to 200 ppm of NH_3_ for 5 cycles at RT and RH of 40%; (**d**) the repeatability of Ti_3_C_2_T_x_ MXene sensor alkalized with 5 M NaOH for 12 h to 200 ppm of NH_3_ for 5 cycles at RT and RH of 40%; (**e**) long-term stability of Ti_3_C_2_T_x_ MXene sensor to 200 ppm of NH_3_ for about 17 days at RT and RH of 40%; (**f**) long-term stability of Ti_3_C_2_T_x_ MXene sensor alkalized with 5 M NaOH for 12 h to 200 ppm of NH_3_ for about 17 days at RT and RH of 40%.

**Figure 5 nanomaterials-14-00680-f005:**
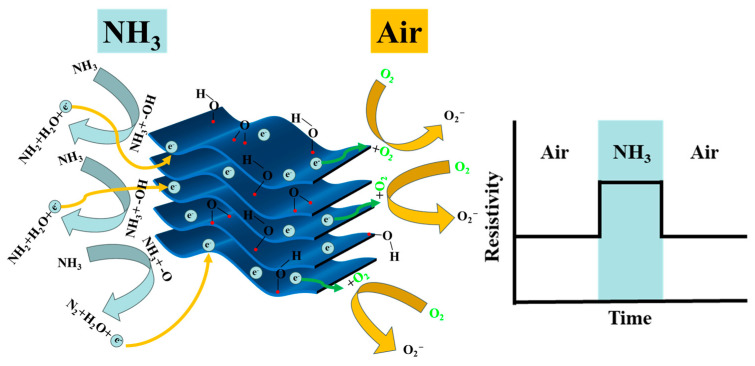
Schematic diagram of the charge transfer mechanism and electrical conductivity changes in NH_3_ and air.

**Figure 6 nanomaterials-14-00680-f006:**
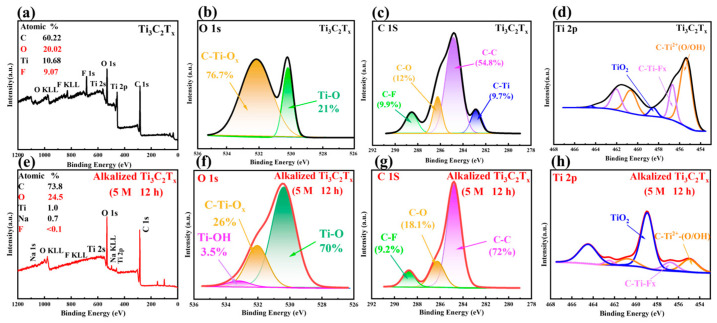
XPS spectra of Ti_3_C_2_T_x_ (**a**–**d**) and Ti_3_C_2_T_x_ alkalized with 5 M NaOH for 12 h (**e**–**h**): (**a**,**e**) Survey XPS spectra; (**b**,**f**) O 1s spectra; (**c**,**g**) C 1s spectra; (**d**,**h**) Ti 2p spectra of Ti_3_C_2_T_x_ and Ti_3_C_2_T_x_ alkalized with 5 M NaOH for 12 h.

**Figure 7 nanomaterials-14-00680-f007:**
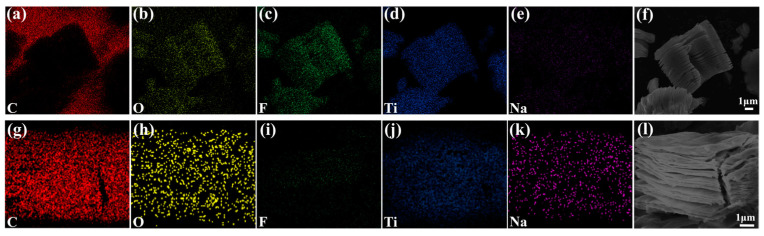
The EDS element mapping scanning images of Ti_3_C_2_T_x_ (**a**–**f**) and alkalized Ti_3_C_2_T_x_ (**g**–**l**).

**Figure 8 nanomaterials-14-00680-f008:**
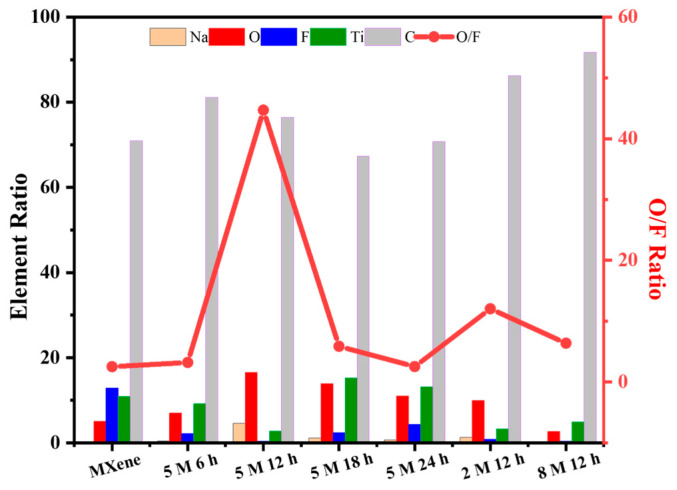
The statistical graph of Na, O, F, Ti, and C element ratio and O/F ratio of Ti_3_C_2_T_x_, alkalized Ti_3_C_2_T_x_ (5 M NaOH for 6, 12, 18, 24 h, 2/5/8 M NaOH for 12 h).

**Table 1 nanomaterials-14-00680-t001:** Comparison of RT NH_3_-sensing properties in the present work and other literature.

Materials	NH_3_ (ppm)	Response (%)	Response/Recovery Time	Ref.
Ti_3_C_2_T_x_	100	0.21	-	[9]
Ti_3_C_2_T_x_	100	0.8	-	[37]
Ti_3_C_2_T_x_	100	0.12	6.8/13.3 min (100 ppm NH_3_)	[38]
Ti_3_C_2_T_x_/Graphene	100	6.77	3/12 min (100 ppm NH_3_)	[39]
3D Ti_3_C_2_T_x_	100	0.8	1.5/1.7 min (5 ppm NH_3_)	[40]
Ti_3_C_2_T_x_/SnO_2_	100	3.1	109/342 s (100 ppm NH_3_)	[41]
MXene/TiO_2_/cellulose	100	6.84	76/62 s (100 ppm NH_3_)	[42]
Alkalized Ti_3_C_2_T_x_	100	7.04	73/38 s (200 ppm NH_3_)	This work
Single-layer Ti_3_C_2_T_x_	500	6.13	45/94 s (25 ppm NH_3_)	[43]
Ti_3_C_2_T_x_ MXene/MoS_2_	500	5.8	-	[44]
Alkalized Ti_3_C_2_T_x_	500	78	-	[31]
Alkalized Ti_3_C_2_T_x_	500	100.3	73/38 s (200 ppm NH_3_)	This work

## Data Availability

All data are contained within the article.

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
