# Peer review of "Sensitivity-Enhanced, Room-Temperature Detection of NH3 with Alkalized Ti3C2Tx MXene"

_nanomaterials, 2024, doi:10.3390/nano14080680_

Round 1
Reviewer 1 Report
Comments and Suggestions for Authors
In this article, authors reported the room temperature NH3 gas sensing performances based on alkaline Ti3C2Tx MXene sensor via NaOH treatment. The manuscript was written well and the results are good in competing with the existence literatures. However, there are some concerns that needs to be clarified before the publication, and here are some of my concerns as given below.
1. What does the exact meaning of the organ like layered structure please explain.
2. Author reported that 0.5 g of prepared Ti3C2Tx MXene was added into a 30 mL 2/5/8 M NaOH solution and then the mixture was allowed to stirred for 6, 12, 18, 24 h, respectively at 99 room temperature and RH of 40%. It means that each M of NaOH was stirred at each different temperature range (6-24 h). But in the results authors explained the XRD results based on time constant with varied molar concentration. Please specify.
3. Please explain that how did the authors estimated the gas ppm concentration in static mode. Explain the detailed method of estimating the ppm levels using the dynamic standard equation step by step, which might be useful for the readers and researchers.
4. Please describe d and included the sensor fabrication details, what is the area of each sensor.
5. How did the authors declared that after alkalization, mainly -O and - OH surface groups remain with -F rich surface terminations being transitioning to -O and -OH rich terminations, please provide sufficient explanation and cite relevant literature since the novel works some time needs refences to strengthen their unique results?
6. It can be seen that with increase in molar concentration the XRD peak intensity and the FWHM also decreases. Please specify and the also mention more reasons for the optimization of 5M/12h. is the best sample.
7. The ppm levels used in this work is relative higher. For the practical applications view this ppm levels are little high enough.
8. Please proof read the entire manuscript to avoid the punctuation marks, space errors, and suffix prefix errors.
9. Please explain the baseline in-indentation (exponential) to the stability test in Fig. 4(c).
10. Please check the figure caption in Figure4. (c and d) little bit confusion with respect to the Figures displayed.
11. Provide error analysis for better accuracy of results and the detection limits of the sensors.
12. Please increase the font sizes of all the images for the better view of the displayed words.
13. Please explain the charge transfer mechanism that corelates with the adsorption/desorption of NH3 molecules, which is much appreciable and highly useful to the areas and researchers if authors explained using the band analysis.
14. Provide few more literatures in the comparison table to further enhance your proposed structural sensing results.
15. Some of the references are not up to the mark, a few of the latest reports are suggested to cite mechanism and 2D materials related discussions, such as Journal of Alloys and Compounds 938 (2023): 168563, ACS Materials Lett. 2023, 5, 10, 2739–2746.
Author Response
Thanks for your comments and suggestion. We greatly appreciate the valuable suggestions and positive comments. All the comments have been seriously considered. Please be noted that reviewer comments are shown in blue type, our responses in black type and the revised portions are marked in red in the manuscript. We hope you are satisfied with the revised manuscript.

Reviewer 2 Report
Comments and Suggestions for Authors
The study adresses to the alkalization of MXene for the room-temperature detection of ammonia. I found work well-performed, however it is neccesary to adress some issues and questions before publication:
1. How does the electrical conductivity of materials changes depending on the alkalization time?
2. Why does the shape of the experimental curves so much differ in the figures 3, 4(a,b,e,f) from that in 4(b,c)?
3. The XPS results should be reconsidered. The deconvolution of XPS spectra is not apropriate:
- Some advises in peak fitting you can find in https://doi.org/10.1116/6.0001975
- The Ti 2p spectra is always split into duplet (Ti 2p 3/2 and Ti 2p 1/2), the separation between 3/2 and 1/2 should be consistent for all components. More detailed information you can find in https://doi.org/10.1016/j.matt.2021.01.015
4. The change of elemental composition depending on the alkalization time seems inconsistent. The F and Ti content decreases and then increases. How can the authors explain this issue?
5. It would be helpful to benchmark the sensor performance with the related sensor materials.
Comments on the Quality of English LanguageMinor editing of manuscript is needed.
There are some missed spaces:
- line 169
examples of bad phrasing:
- line 266, line 316
undeciphered abbreviation:
- line 239
Author Response

(The authors gave the same response as above.)

Reviewer 3 Report
Comments and Suggestions for Authors
In the manuscript, Y. Tan et al. report on preparing organ-like Ti3C2Tx MXene, its characterization, and the fabrication of a gas sensor based on the acquired material. In general, it is a decent study, diversifying the library of gas-sensing materials. Experimental studies are performed on an appropriate scholarly level. However, several omissions make the manuscript premature for publication in Nanomaterials and are to be addressed, requiring major revision. The following points need to be considered by the authors:
1. The authors use the term “organ-like structures” several times throughout the text. It is not well-known by a broad scientific community and, thus, should be defined in the Introduction section
2. More details on performing the deconvolution of the acquired XPS spectra must be provided. See, for instance [10.3390/nano13010023]
3. Furthermore, a more detailed description of the gas sensor setup, such as the material and geometry of the measuring electrodes, the thickness of the deposited MXene layer, etc. should be provided. In its current form, the manuscript does not provide a clear presentation of the developed gas sensor and complicates the evaluation of the performed gas sensing measurements
4. The authors claim that the shift of the (002) peak in the XRD patterns shifts towards lower angles, indicating an increase in the interlay distance due to the extraction of the Al layers. This seems clear but why do other layers not stack back after the removal of the Al ones? Is it related to their functionalization by –F, -O, and –OH groups? It seems reasonable to provide these details for better understanding by the readers
5. What are the factors, which make the MXene layer after 12 h of alkalization exhibit the best gas sensing performance? Why such a drastic decrease in the chemiresistive response is observed when the alkalization time is further increased to 24 h regardless of the concentration of NaOH? At the same time, why does the use of 8 M NaOH solution lead to worse gas sensing performance despite the treatment with 2M and 5M solutions results in MXene having almost the same chemiresistive response? Discussion on these questions should be introduced into the text. The claim “It should be noted that the -OH surface functional terminal groups will be saturated if the concentration of NaOH too high and the alkalization time too long” (lines 295-296) is indistinct and does not provide any explanation.
6. The authors are encouraged to provide the same time range in Figures 4a and 4b cause the current representation is confusing, making one think that the alkalized MXene exhibits higher response and recovery times
7. The provided discussion on the effect of the –F and –OH on the adsorption of the ammonia molecules is ambiguous. Both these groups exhibit a partial negative charge, attracting ammonia molecules, with the –F groups higher partial charge. Thus, functionalization by fluorine is even more favorable for the detection of ammonia, which is reflected by vast reports on sensing ammonia by fluorinated graphene and carbon nanotubes. So the straightforward replacement of the –F groups by –OH groups would not lead to an increase in the number of adsorption sites and the increase in the chemiresistive response. This should be explained. Furthermore, authors should either exclude or clarify the term “strong interaction” since if the interaction is really strong (meaning the adsorption energy is of ~0.3 eV or higher) no recovery of the sensor upon simple purging with air would be indicated.
8. In the case of C 1s deconvolution C-F peak should be discerned in the Ti3C2Tx spectrum around ~288 eV [see 10.1088/1361-6528/28/7/074001]. At the same time, the deconvolution of the C 1s spectrum of the Alkalized Ti3C2Tx must be refined since the CHx/CO peak having FWHM of more than 2 eV is unphysical – unless the structure of the material is totally amorphous [see 10.3390/nano13010023].
9. Apart from the deconvolution of the C 1s spectra, authors must also explain the reason for the drastic increase in the C content complemented with the reduction of the Ti concentration according to the displayed Survey spectra. It does not follow the presented discussion on the alkalization of the MXene.
10. Furthermore, no signs of Na 1s peak around 1070-1080 eV can be distinguished in the displayed Survey XPS spectra. If the Na concentration is around 0.7 at.% as indicated in Figure 5d, sodium cannot play any valuable role in the structure and performance of the formed alkalized MXene, which contradicts the authors’ claims (lines 243-244, for instance). This must be clarified.
11. Figure 7 is somewhat ambiguous. What does “element ratio” mean here? If it refers to atomic concentration – then this graph does not match the XPS data (Figure 5). If it is a ratio between the corresponding elements and some chosen element (as it is displayed for the O/F ratio) – it is totally unclear and should be explained in detail
Comments on the Quality of English LanguageExtensive editing of the English language is required. The text is full of misprints, grammatical errors, etc.
Author Response

(The authors gave the same response as above.)

Reviewer 4 Report
Comments and Suggestions for Authors
The authors presented an article in which it was proposed to carry out alkaline treatment of materials to enhance gas sensitivity to ammonia. The surface chemistry of the processes occurring is discussed. The article is well written, but additions and clarifications are required in various parts of the work. More details below:
1. The authors write that they studied the gas-sensitive properties using the static method. However, graphs of the dynamic response of the sensor signal of the samples to NH3, where the duration of the gas pulses is only ~50 s, are presented (Figure 4d, for example). How was it possible to ventilate an 18 L chamber so quickly? Can you provide more information on how the sensory measurements were carried out?
2. (a) The interpretation of the XPS spectra of Ti2p is not entirely clear. Which peaks in the spectrum belong to the spin-orbit splitting components Ti2p 1/2 and Ti2p 3/2 ? Why is each peak assigned to a separate charge state if the spin-orbit splitting components exist in pairs?
(b) In the survey spectra (Figures 5a,d) Ti3s, Ti3p peaks are marked, but the position of these peaks does not coincide in both spectra (region 200-0 eV). Check this and make the correct peak assignment.
(c) It is necessary to add high resolution spectra for the O1s region, because this presence of surface -O and -OH is discussed in relation to the mechanism of gas sensitivity.
3. Can the specific surface area of the samples, and specifically its increase as a result of alkaline treatment, be studied to support the conclusions drawn?
4. In my opinion, a little explanation is required about the mechanism of the sensor signal to NH3. Is this increase a consequence of ionic (proton) conductivity or electron conductivity? What processes on the surface does this correspond to?
5. Adding cross-sensitivity experiments would be extremely interesting and useful.
4. The authors are encouraged to replace the frequently used phrase “room temperature” with the abbreviation “RT”.
Author Response

(The authors gave the same response as above.)

Round 2
Reviewer 1 Report
Comments and Suggestions for Authors
Authors have answered all my queries and the manuscript in now in good form. Hence manuscript is accepted for the publication.
Author Response
Thanks for your comments and suggestion. We greatly appreciate the valuable suggestions and positive comments. The review's comments are all valuable and very helpful for improving the quality of our manuscript, as well as for the important guiding of our future research.
Reviewer 2 Report
Comments and Suggestions for Authors
The authors addressed most of the comments and significantly improved quality of the manuscript. Nevertheless the deconvolution of XPS Ti 2p spectra is still incorrect:
In the figure 6d,h Ti 2p3/2 fitted by T2+ Ti3+ and TiO2, whereas T2p1/2 component by TiO2-xFx, C-Ti-Fx and TiO2. That is not correct, since any component used to fit a Ti 2p3/2 peak should also be used to fit the Ti 2p1/2 peak. Please, correct the number of components and chemical states in both parts of spectra (Ti 2p1/2 and Ti 2p 3/2).
Author Response
Thanks for your comments and suggestion. We greatly appreciate the valuable suggestions and positive comments. All the comments have been seriously considered. Please be noted that reviewer comments are shown in blue type, our responses in black type and the revised portions are marked in red in the manuscript. We hope you are satisfied with the revised manuscript.
Please see the attachment

Reviewer 3 Report
Comments and Suggestions for Authors
The authors have appropriately addressed all the comments. The manuscript is suitable for publication in the Nanomaterials journal in its revised form
Comments on the Quality of English LanguageMinor editing of the English language required
Author Response
Thanks for your comments and suggestion. We greatly appreciate the valuable suggestions and positive comments. All the comments have been seriously considered. Please be noted that reviewer comments are shown in blue type, our responses in black type and the revised portions are marked in red in the manuscript. We have checked the language of the manuscript carefully according to your suggestions. A native English speaker has helped us to modify the English language. We hope you are satisfied with the revised manuscript.
Please see the attachment

Reviewer 4 Report
Comments and Suggestions for Authors
All comments have been taken into account, the article can be published in its presented form.
Author Response

(The authors gave the same response as above.)
